# gRNA Design: How Its Evolution Impacted on CRISPR/Cas9 Systems Refinement

**DOI:** 10.3390/biom13121698

**Published:** 2023-11-24

**Authors:** Cristofer Motoche-Monar, Julián E. Ordoñez, Oscar Chang, Fernando A. Gonzales-Zubiate

**Affiliations:** 1School of Biological Sciences and Engineering, Yachay Tech University, Urcuquí 100119, Ecuador; 2Departamento de Electrónica, Universidad Simon Bolivar, Caracas 1080, Venezuela; 3MIND Research Group, Model Intelligent Networks Development, Urcuquí 100119, Ecuador

**Keywords:** CRISPR/Cas9, machine learning, gRNA, neural networks, deep learning

## Abstract

Over the past decade, genetic engineering has witnessed a revolution with the emergence of a relatively new genetic editing tool based on RNA-guided nucleases: the CRISPR/Cas9 system. Since the first report in 1987 and characterization in 2007 as a bacterial defense mechanism, this system has garnered immense interest and research attention. CRISPR systems provide immunity to bacteria against invading genetic material; however, with specific modifications in sequence and structure, it becomes a precise editing system capable of modifying the genomes of a wide range of organisms. The refinement of these modifications encompasses diverse approaches, including the development of more accurate nucleases, understanding of the cellular context and epigenetic conditions, and the re-designing guide RNAs (gRNAs). Considering the critical importance of the correct performance of CRISPR/Cas9 systems, our scope will emphasize the latter approach. Hence, we present an overview of the past and the most recent guide RNA web-based design tools, highlighting the evolution of their computational architecture and gRNA characteristics over the years. Our study explains computational approaches that use machine learning techniques, neural networks, and gRNA/target interactions data to enable predictions and classifications. This review could open the door to a dynamic community that uses up-to-date algorithms to optimize and create promising gRNAs, suitable for modern CRISPR/Cas9 engineering.

## 1. Introduction

Historically, biotechnology has undergone remarkable advancements and refinements, leading to significant improvements in its methodologies and outcomes [1]. Notably, great progress has been made in the purification, amplification, and editing of genomes. In the context of genome editing and gene insertion, biotechnology has witnessed groundbreaking developments. Various tools and technologies have been devised to manipulate and modify genetic material with unprecedented precision. The advent of techniques such as CRISPR-Cas9 has revolutionized the field by enabling targeted and efficient gene editing, paving the way for potential treatments of genetic disorders and the creation of genetically modified organisms [2,3]. Concurrently, the field of machine learning has been revolutionized by neural networks and deep learning algorithms, facilitating the extraction of valuable insights and identification of patterns in intricate datasets across diverse research areas [4,5,6]. In this review, we aim to explore the synergistic potential of the fusion of both cutting-edge technologies, CRISPR and machine learning, specifically focusing on the design evolution of guide RNAs (gRNAs) for CRISPR/Cas9-mediated gene editing.

### 1.1. CRISPR/Cas: From a Bacterial Defense to a Genetic Engineering Tool

The first contact with this system occurred when Ishino et al. [7] described the nucleotide sequence of the *iap* gene in *Escherichia coli* in 1987. Until those days, scientists had purely basic and trivial knowledge about this system, and they related it as a cluster of repeated sequences solely spaced by different sequences. In 2007, it was proved that CRISPR/Cas is a defensive bacterial, immunity-providing system against aggressive foreign genetic material such as invading plasmids or bacteriophage DNA [8]. Figure 1 shows the CRISPR locus present in CRISPR-harboring bacteria. The CRISPR array is formed by an AT-rich leader sequence containing promoter sequences; integrated foreign sequences known as spacers; and palindromic repeats serving as spacer separators [9,10,11,12]. Even more, The CRISPR-associated (Cas) proteins are translated from the Cas cluster that is commonly surrounding the neighborhood of the CRISPR array [9].

To date, this defense system must complete three stages to provide immunity in the host cell: adaptation, expression, and interference stages [9,13]. The adaptation stage starts as a response to viral infection. A host Cas1-Cas2 multimeric protein complex recognizes the foreign DNA, cuts a specific sequence known as a protospacer, and integrates it into the CRISPR array [10,12]. This array is transcribed and processed in the expression stage to produce the crRNA:tracrRNA complex. Next, this complex is bounded to the Cas9 protein in the interference stage, assembling the Cas9:RNA structure [14,15], which performs the target recognition and target degradation activity [13], incapacitating the bacteriophage from damaging the host (see Figure 2). Protospacer-adjacent motifs (PAMs) located in the non-target DNA strand, adjacent to the target sequence, are recognized by the Cas9 to initiate the cleavage process [14,16,17]. In 2012, the type II CRISPR/Cas9 system was reprogrammed by its pioneers so it could be used as genome editing machinery [3,18]. This reprogramming involved the substitution of the crRNA:tracrRNA complex by a synthetic single guide RNA (sgRNA) [18], simplifying the whole system (Figure 3). The term sgRNA (single guide RNA) is commonly interchangeable with gRNA (guide RNA). For agility and practical concerns, gRNA will be used in this review.

### 1.2. gRNAs and CRISPR On-and-Off Targets

gRNAs and Cas proteins are the foundation for mediating the desired cuts in the gene of interest. Therefore, we must be familiar with some characteristic features of gRNAs. Firstly, the common total number of nucleotides that constitute the gRNA recognition site is approximately 20 nucleotides [18,19,20,21], mostly sufficient to have precise target recognition. The seed sequence at the 3′ side of the recognition site plays a major role in the target recognition specificity of the Cas9:gRNA. The specificity is significantly diminished when there are two or more mismatched nucleotides between the gRNA’s seed sequence and the target sequence [17,19]. Mismatches in the PAM-distal positions also reduce the specificity of the Cas9:gRNA, but they are much more tolerated than PAM-surrounding mismatches [22,23].

Distinct interactions between the target sequence and the gRNA:Cas9 complex can be categorized into two types: off- and on-target bindings. In this context, on-target refers to the ideal hybridization between the complex and the target, and off-target bindings are understood as the hybridization between undesired DNA sequences and the complex. A high similarity between the gRNA sequence and undesired, non-targeting sequences leads to an elevated percentage of off-target bindings. These off-target bindings can be classified based on distinct genetic events. Manghwar, Zhang, and Niu [24,25,26] delineated three types of off-targets: two out of three refer to bulge-based occurrences (i.e., gaps in the DNA/RNA hybridization), and the remainder refers to simple mismatches. Conversely, Borrelli et al. [27] have simplified the classification into two general types, which describe if the off-target is between sequences with high similarity, or between random, non-similar sequences. Off-target bindings and their consequential undesirable effects are inherent to CRISPR experiments. Consequently, numerous strategies have been devised to minimize off-target activity associated with the CRISPR-Cas9 system. These strategies encompass diverse methodologies, including modification of the Cas9 structure, meticulous titration of Cas9 and gRNA delivery, and alterations of the gRNA ribonucleotide structure [21,24,28].

Regarding alterations in the ribonucleotide structure, significant in silico advancements for gRNA design aimed to enhance the performance of the CRISPR/Cas9 system. Strategies such as adding nucleotides at the 5′ end to differentiate between off-target and on-target sites [19], extending the tracrRNA-fused portion for partial target cleavage improvement [22], and strategically positioning specific nucleotides near the PAM region, middle, or end of the sequence for gRNA stability [20,29,30] have been explored.

Up to now, it has been reported that off-target bindings occur even when there are various mismatches between the gRNA and the off-target site [22,31,32]. Machine learning and deep learning are powerful methods that combine informatics and statistics, and have been successfully used in the design of gRNAs [33]. These methods focus their architectures on predicting or classifying off-target and on-target bindings. The results thus generated only differ in principle; on-target models focus on predicting the effectiveness of the gRNA in cutting a specific target gene. These models identify gRNAs that effectively target a desired gene and are often used in gene editing applications. On the other hand, off-target prediction models focus on identifying potential unintended effects of a gRNA in cutting other genes in the genome that are not premeditated targets.

Furthermore, the correct computational and biological prediction of gRNAs hinges upon other key factors including the correct selection of the promoter sequence for the transcription of the gRNA [34], the genetic context of the target (e.g., how accessible the target is for the Cas9), and the datasets used in the distinct algorithms. Considering the critical significance of gRNA in Cas9 protein-mediated recognition and nuclease activity, substantial endeavors have been dedicated to designing, optimizing, and developing highly effective gRNAs. This review presents a meticulous description of the computational approaches employed in gRNA design, elucidating the intricate intricacies and underlying mechanisms. Through an exploration of these approaches, readers can attain a comprehensive understanding of the evolutionary progression in time and refinement in gRNA design, encompassing both theoretical foundations and practical implementation.

## 2. Machine Learning in gRNA Design

Machine learning is a branch of artificial intelligence [35] that includes algorithms and mathematical models that allow computers to learn from data without being explicitly programmed for each task. The machine learning algorithms follow some steps, starting with data processing, feature extraction, training, and classification or prediction [36,37]. For data processing, the input data are DNA or RNA sequences that require processing to be transformed into a numerical sequence or a format that the computers can work with. There are three general approaches to encode the sequence into a numeric representation: ordinal encoding, one-hot encoding, and k-mer word embedding [38].

In ordinal encoding, each nucleotide is assigned a real value, while in one-hot encoding, a binary vector is assigned [39]. One-hot encoding does not capture any information about the relationship between nucleotides or the context in which they appear, while k-mer word embedding does. However, k-mer word embeddings do not preserve the original sequence of nucleotides and can be sensitive to rare or unseen sequences [40]. Once the data are processed, they undergo feature extraction, which involves selecting and transforming the essential characteristics or patterns of the raw data [41]. Common extracted features from gRNAs are positions of nucleotides, secondary structures of gRNAs, GC count, nucleotide content, nucleotides appearing in a k-mer guide, nucleotides adjacent to the PAM, presence of DNA motifs [42,43], etc. The major features of off-target prediction are the number, composition, and combination of mismatches [22]. During training, a machine learning model searches for patterns in the data. This process also requires setting the hyperparameters, which are variables that control the algorithm’s behavior during learning [36]. For example, the number of layers, neurons, or the type of optimizer are types of these hyperparameters. The manipulation of these items is essential for enhancing the evaluation metrics of the model, so association between biologists and computists is always useful.

Machine learning models can output a sequence for prediction tasks or a categorical label for classification tasks. Classification models can be applied to classify between two options; in our context, this would be whether a given RNA sequence has potential on-target or off-target activity. Prediction models of CRISPR can forecast the effectiveness of a particular CRISPR-Cas system on a given target sequence [37]. The main machine learning algorithms for predicting or classifying DNA sequences are linear regression algorithms, logistic regression, Decision Trees, Random Forest, and Support Vector Machines (SVMs). Neural Networks (NNs) are another algorithm of artificial intelligence, but due to their complexity, these will be explained in another section.

Linear regression (LR) is a learning algorithm used for prediction tasks. Linear regression models fit a linear function between the dependent variable and the independent variables [36]. Some of the studies using this algorithm are CRISPRScan [44] and CRISPRater [45]. A decision function can be added to a regression model to separate the data into two groups. In Logistic Regression (LG), a linear function is transformed through a sigmoid function to produce a probability value between 0 and 1. Some models using this algorithm are Broad GPP [29] and SCC [46].

Decision Trees (DTs) are a supervised learning method used for classification and prediction tasks in DNA analysis. The algorithm builds a tree-like model of decisions and their possible consequences, where each node represents a feature of the DNA sequence and each branch represents a possible outcome based on that feature. The algorithm recursively splits the data into subsets based on the most informative features until a stopping criterion is met. An algorithm derived from decision trees is the Random Forest (RF) [35]. Random Forest (RF) solves the decision tree by using an ensemble process to build multiple decision trees under randomly selected subsets of the data and features, then combines the results of these trees until a good classification or prediction result is obtained [36]. Random forest has been widely used in the prediction of DNA-binding proteins, microarray data analysis, and regulatory element prediction [47]. Examples of these algorithms are CRISTA [48], Elevation [49], and CHANGE-seq [50].

Support Vector Machines (SVMs) are learning algorithms for classification and prediction tasks. The goal of SVM is to separate the data into at least two distinct classes, similar to Linear Regression finding the best line that fits the data [36]. SVM can handle complex data distributions and cases where the classes are not entirely separable by a straight line. An SVM involves two separate steps: feature extraction and training. In these steps, the SVM algorithm learns how to separate the different categories by finding an optimal boundary. Examples of studies using SVM are WU-CRISPR [51], SgRNAScorer [52], Azimuth [53], ge-CRISPR [54], and Predict CRISPR [55].

Despite the several models and their different architectures, the mechanism behind them is very similar, and the models differ mainly in the way they are trained (Figure 4). In the case of machine learning algorithms, we highlight the RF and SVM algorithms since they are the most representative and have a more complex mechanism compared to the LR and LG algorithms. In the case of RF, Figure 4A shows that training is performed on each decision tree. This mechanism is similar to that of SVM (Figure 4B), but in the latter, the algorithm is trained iteratively on the data and produces a single output. In the case of NNs, their mechanism is much more complex than in the case of machine learning algorithms, but as an introduction, NNs have layers of neurons that perform a specific computation and process different types of data. In the case of Recurrent Neural Networks (RNNs), the layer of neurons that defines them is the Hidden Layer (Figure 4C), and in the case of Convolutional Neural Networks (CNNs), it is the Convolutional Layer (Figure 4D). In the next section, we describe in depth NNs and the mechanisms that govern them.

## 3. Advances in Neural Networks for gRNA Design

NN represents a branch of machine learning and artificial intelligence [35] inspired by the complex organization and functioning of the human brain. NNs process input data and generate predictions by combining interconnected artificial neurons arranged in layers. These models employ mathematical operations, activation functions, and adjustable weights to transmit and transform information across the network [36], enabling them to discern patterns in the data and process complex information. The output layer of an NN produces predictions or classifications based on the learned information. Through the iterative backpropagation process, where inputs are repeatedly fed into the network, and their outputs are compared to desired results, the NN adjusts its parameters (weights and biases) to minimize the discrepancy [35]. This iterative refinement continues until the network converges upon optimized weights that minimize the loss function.

In gRNA prediction, NN has emerged as an indispensable tool for evaluating gRNA effectiveness [56]. NN models are trained using vast datasets, enabling them to extract complicated patterns from the data, facilitating precise gRNA prediction. NN learns to predict gRNA by iteratively adjusting its parameters, namely the weights and biases of its neurons, through a process known as backpropagation. During backpropagation, the network receives input data, compares its output to the desired output, and updates the parameters to minimize the discrepancy between them [35]. This iterative process is repeated multiple times until the network converges to optimal weights that minimize the loss function. As a result, the NN acquires the ability to make accurate predictions or classify inputs based on the discerned patterns within the training data.

The architecture of an NN plays a key role in its performance, with Feed Forward Neural Networks (FNNs), CNNs, and RNNs being prevalent choices designed for sequences [34,57]. The absence of feedback connections between neurons characterizes the FNN architecture. The inputs are propagated forward through the network, with each neuron receiving inputs from the previous layer. This flow of information occurs in a single direction, without any loops or cycles. CNNs, initially designed for image processing, have been adapted for gRNA prediction tasks using the 1D-CNN architecture [58]. They excel in extracting features and training simultaneously and have become integral to gRNA prediction models [41]. Conversely, RNNs are specifically tailored for sequential data and can capture dependencies in gRNA sequences due to their recurrent connections, so the information flows in two directions [35].

The utilization of NN in CRISPR prediction has witnessed substantial progress over time, driven by advancements in architectural designs and empirical investigations. Researchers have explored various NN configurations and conducted experimental studies to enhance the accuracy and efficacy of gRNA predictions. In the following sections, we shall elucidate the evolutionary trajectory of these computational approaches, delineating the architectural refinements and pivotal experimental discoveries that have propelled the advancement of gRNA design.

## 4. Evaluating Model Metrics

Evaluation metrics are important measures used to assess the performance of predictive models. In CRISPR gRNA prediction, the most commonly used metrics are the Spearman correlation coefficient, the AUROC (Area Under the Receiver Operating Characteristic) curve, and the accuracy. The latter is a commonly used metric to evaluate classification models. It measures the proportion of correctly classified instances over the total number of instances. The accuracy provides a single scalar value representing the prediction’s overall correctness [35].

The Spearman coefficient measures the relationship between two variables and is more appropriate than Pearson’s correlation [59]. In experimental data analysis, high Spearman coefficients indicate a strong correlation between the predicted and experimental rankings and consequently high moel accuracy [60]. In the context of CRISPR gRNA prediction, the Spearman correlation coefficient serves as a robust benchmark to evaluate the predictive performance of a tool. This coefficient has been widely employed in the existing literature as a reliable measure of the correlation between predicted and experimental rankings [34,48,49,61,62,63].

The AUROC curve evaluates the ability of a model to distinguish between positive and negative samples. It measures how well the predictions are on balanced or unbalanced data, with a higher AUROC score indicating better performance. AUPRC (Area Under the Precision-Recall Curve) is a useful performance metric, particularly in scenarios involving imbalanced data where identifying positive instances is extremely important. A higher AUPRC score signifies superior classifier performance, with a score of 1 indicating a perfect classifier and 0 representing poor performance [64]. Compared to AUROC, AUPRC is typically smaller in magnitude and provides a more suitable measure for off-target prediction. This is crucial in clinical applications where false negatives have far more adverse effects than false positives [65].

While all these metrics are used to assess a model’s performance, they are not directly comparable because they measure different aspects of a model. Additionally, most studies use different datasets, further complicating a meaningful comparison. However, these metrics analyze each model’s individual performance and give insight into how models have evolved. If needed, these metrics will be inspected in the subsequent section to assess the temporal evolution and advancements in the models. By considering these metrics, we can gain insight into the extent of changes and improvements observed over time.

## 5. Reaching Efficiency: A Journey of Optimization

Since the beginning of gRNA design algorithms, scientists have widely used these programs to find the ideal gRNA for their experiments. The efficiency of these programs is important for research, and it can be measured according to the evaluation metric previously presented. For every tool described below, metrics and details are present in Table 1 and Table 2. Furthermore, in Figure 5 and Figure 6 we present a timeline with a description of each tool, their computational architecture, and specialization.

In 2013, Hsu et al. [22] published the first web-based off-target predictor, CRISPRtool, also known as **MIT CRISPR Design**. Through a series of experiments in human cells, they built an enormous database of the activity of distinct gRNAs and unveiled diverse features that will help to reduce off-target cuts. All these approaches were implemented in a conventional algorithm that supports their tool. Thus, in 2014, the CRISPRtool helped to design the gRNAs targeting two tumor suppressor genes and one oncogene, mutating them directly for mouse lung cancer; their transient transfection reached a maximum of 44% of indels [78]. Almost the same results were obtained in the work performed by Xue et al. under the same conditions, but for liver cancer in mice [79]. Evidently, MIT CRISPR Design was focused on predicting off-target sites and designing gRNAs for Cas9 proteins. Conversely, in April 2014, Montague et al. [80] developed **CHOPCHOP**, a web tool that directly competes with MIT CRISPR Design, adding support for TALENs and CRISPR/Cas9 genome engineering. CHOPCHOP can accept diverse inputs and utilizes a strict algorithm for alignments, named Bowtie [81], to predict off-target sites within the genome accurately. All potential options are then presented in an interactive graphical interface. The currently available version of CHOPCHOP (https://chopchop.cbu.uib.no; accessed on 30 April 2023) supports gRNAs designed for other CRISPR nucleases, such as Cpf1 or Cas13.

The previous tools were developed by using gRNA datasets for mammals. Thus, in the middle of 2014, Lei et al. [82] launched **CRISPR-P** to design specific gRNAs for gene targeting in more than 20 plant species. The computational model supporting this tool is based on BLASTN, which compares the gRNA sequence with all possible off-target sites in the genome. To perform the predictions, users must select the plant of interest, upload the FASTA sequence of the targeting gene, and select diverse parameters of their convenience; outcomes are presented in lists or graphical representations. The web-based tool (http://crispr.hzau.edu.cn/CRISPR; accessed on 30 April 2023) is available and receiving constant updates.

Approaching September 2014, Doench et al. [29] launched the **Broad GPP Designer**, currently relaunched and updated as CRISPick. This on-target prediction engine first used a learning algorithm, in this case, a logistic regression algorithm to create a robust model. As long as logistic regression is intended to produce a binary output, this model predicts whether or not a gRNA effectively inactivates a gene. The authors used their own dataset of 1841 gRNAs to train the model; this dataset included information about the sequence of each gRNA, as well as whether or not the gRNA was effective in inactivating a gene. Finally, in practice, users can input a target DNA sequence and select a PAM sequence, and the tool will output a list of optimized gRNAs that target that sequence. Research about genome editing in the parasite *Leishmania donovani* was performed using CRISPR/Cas9 [83]. Here, the authors compared their hand-crafted gRNA with a set of gRNAs suggested by the Broad GPP Designer for the gene of interest, resulting in a low score for the hand-crafted gRNA according to the web tool output. Furthermore, the concerned tool helped to create a robust, highly efficient protocol to mediate genome editing in *Caenorhabditis elegans* regardless of possible low-efficiency gRNAs, permitting the use of a wider variety of gRNAs [84].

The off-target prediction was purely based on computational programs until the end of 2014 when Tsai et al. [85] presented **GUIDE-seq**, a laboratory technique that enables genome-wide profiling of off-target cleavage by Cas9 nucleases. Together with the technique, the authors discovered that the MIT CRISPR Design tool did not identify the vast majority of off-target sites found by GUIDE-seq due to very limited parameters implemented in the algorithm.

In June 2015, Xu et al. [46] launched the linear regression-based **Spacer Scoring for CRISPR** (SSC) tool. They showed distinct problems with the gRNA sequence features unveiled to that date and the lack of a model that predicts gRNAs for CRISPRi/genome-wide functional screens. Thus, the SSC tool agglutinates two computational models, based on linear regression, for predicting gRNA efficiency in CRISPR-based screens. These models were designed to correspond to CRISPR knockout and CRISPRi/a studies, respectively, and were validated through multiple tests on independent designs, various cell types, and different growth selections. These tests used the AUROC benchmark, for which the MIT CRISPR Design tool was outperformed by the SSC tool. Hence, Radzisheuskaya et al. [86] utilized this tool to confirm that employing the correct gRNAs explicitly designed for functional genome screens will improve efficiency. In other words, in the case of CRISPRi, if the gene transcription start site (TTS) is targeted and the highest-scored gRNA is used, the efficiency will increase, showing better phenotype-based screens.

Plant gRNA prospects and their characteristics partially differ from gRNAs designed for mammals or bacterial cells. For instance, nucleotide preferences in the recognition sequence are not seen for plants, unlike for mammals [87]. Besides linear or logistic regression-based machine learning methods in the last tools, **WU-CRISPR** and **SgRNAScorer** [51,52] implemented the Support Vector Machine (SVM) framework for gRNA design in November 2015 and July 2015, respectively. The SgRNAScorer model leveraged SVM to capture complex associations between gRNAs exhibiting high and low activity. The authors used their own experimental data to train their model. On the other hand, for WU-CRISPR model training, the authors used the MIT Design dataset [22], fusing the SVM with novel features. WU-CRISPR faced the SSC [46], SgRNAScorer [52], and GPP CRISPR designer tool [29], demonstrating that the WU-CRISPR model has a better performance through the respective benchmark. Part of its success is due to considering the presence of contiguous repetitive sequences, or the impact of secondary structures (such as hairpins, etc.) formed in the guide sequence occasioned by self-folding free energy [87]. Mutagenesis experiments in rice and cotton employed the SgRNAScorer for gRNA design. In cotton experiments [88], the SgRNAScorer designed 82 distinct gRNAs to target a GFP gene in a transgenic cotton genome, selecting only three significantly different gRNAs in the scoring value. They found that the mutagenesis efficiency varied inconsistently, suggesting that these gRNA prospects lack robust biological and computational basis. Interestingly, after these results they decided to use WU-CRISPR, obtaining only 13 gRNAs for their gene, validating experimentally computational benchmarks. Analogously, in rice experiments, Baysal et al. [89] used the SgRNAScorer to design two gRNAs for a gene of interest. Unfortunately and inconsistently, in these experiments, the gRNA with the highest score showed no mutagenesis activity, whereas the lowest-scored gRNA positively did. WU-CRISPR is available at (https://crisprdb.org/wu-crispr-website; accessed on 30 April 2023), where we can select designed gRNAs for mouse or human genes.

A linear regression-based model was introduced by Moreno-Mateos et al. [44] in August 2015 with **CRISPRscan**. The model was created on the basis of results obtained from analyzing the stability and mutagenic activity of 1280 gRNAs targeting 128 genes. The authors discovered that guanine enrichment and adenine depletion are major determinants of gRNA stability and activity, but impressively, they found that at least in their experimentally validated gRNAs, treating the folding free energy does not contribute to any improvement to the model. However, Thyme et al. [90] explain that hairpin formation can reduce gRNA efficiency, and many web-based tools before 2016 ignored this critical factor. CRISPRscan was not the exception, but it presented a lower hairpin formation fraction compared with their contenders. In 2016, research about genome modification in hematopoietic stem/progenitor cells (HSPCs) was significantly improved by Gunry et al. [91]. They targeted the *CD45* gene in human HL-60 cells with three distinct gRNAs designed with CRISPRscan. Here, high mutagenesis percentages were obtained, touching almost 75% of indels, which classifies CRISPRscan as a high-fidelity gRNA design tool, at least in mammal cells. The CRISPRscan web page is available and implements all the approaches mentioned here (http://www.crisprscan.org; accessed on 30 April 2023).

At the start of 2016, Doench et al. [53] launched the **Azimuth** project, an essential update of the Broad GPP Designer. In the Azimuth web page, or as a Python code, the authors provide two score-based parameters for accurately discriminating potential on-target and off-target sites: **Rule Set 2**, and the **CFD score**. Rule Set 2 was developed by doubling the size of the gRNA activity data set and incorporating gradient-boosted regression trees (GBRTs), an ML algorithm that combines the predictions of multiple decision trees to make more accurate predictions. Even more, Rule Set 2 has such a performance that extends to accurately predicting on CRISPRi/a screens. On the other hand, the CFD score was validated using GUIDE-seq data [85] showing the best Pearson correlation coefficient between experimental and predicted activity. The Azimuth scores integrate biochemical and thermodynamic sequence features regarding the secondary structure formation, a characteristic missing in the Broad GPP. To date, the Azimuth web page is no longer available. For further incorporation in tools or pipelines, only Python code is available in the supplementary information attached to the published article.

As explained throughout this section, the off-target predictor, MIT Design, by Hsu et al. [22] suffered from many weaknesses, invoking the necessity of potent tools to predict off-target sites, each one surpassing the others. In June 2016, Haeussler et al. [66] launched the off-target predictor **CRISPOR**, principally designed to assist with guide selection in 120 genomes, including plants and many other organisms. The algorithm is powered by the BWA (Burrows–Wheeler Alignment) sequence search algorithm [92] to perform the corresponding alignments to locate possible off-target sites. CRISPOR’s predicted gRNAs avoid using extremely GC-rich sequences, and the tool treats >4 mismatches much better than the MIT Design. These patterns found by analyzing eight large datasets of off-target sites deliver an improved fidelity on CRISPOR prediction. Finally, the CRISPOR web page (http://crispor.gi.ucsc.edu/; accessed on 30 April 2023) offers pre-calculated gRNAs for all human exons on the UCSC Genome Browser tracks, CG content warnings, PCR primers, etc. In laboratory experiments, mutagenesis and gene knock-out research in the hexaploid *Camelina sativa* [93] employed the CRISPOR tool to design desired and exclude undesired gRNAs for targeting the microsomal oleate desaturase (*FAD2*) gene, whose knock-out leads to an accumulation of oleic acid in this plant. They selected two gRNAs, from which the second one harbors sequence features described by CRISPOR to improve the mutagenesis efficiency.

Looking back on the SgRNAScorer, Chari et al. [52] structured this tool to analyze gRNA sets of high and low activity for two orthologs of the Cas9 protein. For each ortholog, a separate SVM model was created. In February 2017, the same team founded the **sgRNA Scorer 2.0** [71], which inversely creates just one SVM model for both Cas9 orthologs by merging all high- and low-activity gRNAs in one set.

With this, they aimed to design a model that predicts efficient gRNAs for distinct Cas9 orthologs, and even for Cpf1 proteins, a non-Cas9 system. The web page is still available (https://frederick.cancer.gov/resources/repositories/sgrnascorer; accessed on 30 April 2023), and users can select the CRISPR system in usage, PAM sequence, and upload gene FASTA sequences for analysis. Despite this tool being trained with a dataset of gRNAs targeting human cell genes, Shen et al. [94] used this tool to design 81 gRNAs targeting virulent *Klebsiella* phage genes. As expected, due to the cellular context, sgRNA Scorer 2.0 did not discriminate correctly between high- and low-activity gRNAs.

In 2016, the research carried out for CRISPOR’s feature incorporation caused inconsistencies with the research by Abadi et al. [48]. The latter group launched in 2017 a new predictor known as **CRISTA** (CRISPR Target Assessment), based on a regression model using the Random Forest algorithm. Their primary purpose was not to design a model for exclusively predicting gRNA on-target efficiency or potential off-target sites but to assess the cleavage efficacy of a particular genomic target by a specific gRNA. CRISTA included a treatment for DNA/RNA “bulges” in their algorithm, which can be understood as gaps in the gRNA/target hybridization. CRISPOR noticed these bulges, but their analysis suggested no need for treating these gaps, disfavoring this tool for missing this important feature. CRISTA finally proved experimentally that bulges are important determinants in gRNA design, and considered the necessity of dealing with the formation of secondary structures inherent to RNA sequences by their learning model. Furthermore, the DNA enthalpy, geometry, and the target location (chromosome number and distance from telomere and centromere) are some additional features inserted in the algorithm. In contrast to many other predicting tools, the CRISTA training dataset does not discard uncleaved sites (i.e., targeted sequences with no gRNA activity), helping to avoid the design of identical zero-activity gRNAs. Finally, the CRISTA web page (crista.tau.ac.il) and downloadable Python code are available.

In January 2018, Listgarten and colleagues developed the **Elevation** tool [49], an off-target predictor algorithm that aims to complement the Azimuth model, changing the architecture for a two-layer stacked regression model and a gradient-boosted regression tree (GBRT), where the first layer is intended to learn to predict unique mismatches in the gRNA-target duplexes. The second layer learns to predict various mismatches, yielding a score for potential off-target sites. Finally, they provide a minimalist web page (https://crispr.ml/; accessed on 30 April 2023) that users can select if they need to search off- and on-target sites.

The CRISPR/Cas9 genome editing system left the scientific community with gigantic expectations. The promise of flawless gene knock-out, knock-in, or functional screens must be accomplished. In June 2018, Chuai et al. [67] finally included the use of deep neural network (DNN) approaches for predicting and designing gRNAs into their novel tool, **DeepCRISPR**. In parallel with CRISTA, DeepCRISPR seeks to predict both functional on-target gRNAs and avoid those with a propensity to increase off-target cuts. To achieve this purpose, the authors designed an architecture with three fundamental networks; the main one can be understood as the pre-training network (known as “parent network” by the authors), which will recognize various features of gRNAs using as input ∼0.68 billion gRNA sequences targeting coding and non-coding human genes. This first network is based on a DCDNN (deep convolutionary denoising neural network) architecture to tolerate huge quantities of noise in the input. The next two CNNs receive as input the pre-training network output. These latter networks are trained using well-known, experimentally validated gRNAs with on-target and off-target activity, extracting all the distinctive features characterizing these sequences for further integration in the predictive capacity of the tool. In 2020, accordingly to the pre-training DeepCRISPR dataset based on human exons and intron genes, the tool helped to predict the off-target activity of gRNAs designed by Mintz et al. [95] that initially targets the *PARP1* gene, for its inhibition in triple-negative breast cancer (TNBC) cells, highlighting the importance of CRISPR/Cas9 systems and high-performance gRNAs in preclinical studies. Unfortunately, the web page the authors provided is no longer available.

Three months after the DeepCRISPR launch, Lin et al. [39] developed a tool that exclusively predicts off-target sites with a DNN (Deep Neural Network) framework. They named their tool **CNN_std**, in which they adapted the biological ribonucleotide sequence of the gRNA for the computational environment in a matrix with a size of 4 × 23, representing the four nucleotides and the 20 nt recognition sequence plus the 3 nt PAM sequence. This matrix has the correct format to be an input for the CNN. Furthermore, Lin et al. utilized the CRISPOR dataset [66] and GUIDE-seq data [85] for training, validating, testing, and comparing CNN_std against previous off-target prediction tools such as the CFD score or the MIT CRISPR design tool, outperforming all these and other machine learning-based tools and obtaining an AUROC of 0.972. The authors do not provide a web page integrating this tool, but a Python code is provided to use the tool (https://github.com/MichaelLinn/off_target_prediction; accessed on 30 April 2023).

Undoubtedly, the CRISPR/Cas9 systems had an enormous refinement with the introduction of CNN. Unluckily, DeepCRISPR and CNN_std implemented algorithms and architectures that neglected most of the biological features underlying the gRNA feasible design, thus missing characteristics proved to be crucial for this objective. Furthermore, sgRNA Scorer, the first and second version [52,71], used an algorithm trained with datasets obtained from diverse Cas9 orthologs, then being capable of predicting under a more comprehensive array of RGNs (RNA-guided nucleases). In September 2019, Wang et al. [34] compared the predicting performance of an RNN and a conventional CNN. They found that RNN outperforms CNN and other machine learning algorithms. The dataset used for training, validation, and testing is based on their own experiments in human cells, emphasizing the use of three Cas9 orthologs: WT-SpCas9 (wild-type *Streptococcus pyogenes* Cas9), eSpCas9 (enhanced), and SpCas9-HF1 (High Fidelity). Furthermore, to remedy the previous nonexistence of biological feature treatment in DNN models, this RNN was trained with features such as sequence secondary structure formation and their stem-loops, GC content, or the contiguous repetitive sequences that were first implemented in WU-CRIRPR [51]. Lastly, Wang and his team launched **DeepHF**, a tool addressing all the concerns mentioned earlier. DeepHF was used in experiments premeditated to knock out an apoptosis-inducing gene in mice, *Htra2*, whose translated protein is found in high concentrations in neomycin-treated cochleae, one of the causes of deafness. The team designed three gRNAs targeting the *Htra2* gene, obtaining 87.27 % of indels in the *Htra2* gene for the highest-scored gRNA [96]. The authors provide a Python code for integration in new tools (https://github.com/izhangcd/DeepHF; accessed on 30 April 2023), and a module-based web page (http://www.DeepHF.com/; accessed on 30 April 2023) that lets the users design and view the characteristics of gRNAs with an input sequence.

Notwithstanding the boom of deep learning-based pipelines in gRNA design tools, Muhammad et al. [75] were uncomfortable using DNNs for gRNA design. Despite the visible characteristics and performance obtained with these models (CNN or RNN), it is tough to interpret their methods and results. Even more, it has been proven that conventional, simpler algorithms can perform the same work done by DNNs [97]. With this, in June 2020, Muhammad et al. launched the on-target **CRISPRpred(SEQ)** tool, whose SVM-based architecture was trained with the same training dataset for DeepCRISPR while mixing biological gRNA sequence features. In most of the benchmarks, CRISPRpred(SEQ) outperformed DeepCRISPR. On the other hand, CRISPRpred(SEQ) challenged DeepHF using the dataset generated by the latter; unluckily, the machine learning-based tool did not surpass DeepHF due to needing more specific tuning against DeepHF. The authors do not provide an interactive web page. Instead, Python code is provided in which we can train the model with the user’s new data (https://github.com/Rafid013/CRISPRpredSEQ; accessed on 30 April 2023).

Another approach to achieve the desired interpretability in DNN is presented by Xiao et al. [62]. Firstly, they provide a categorization of the existing DNN, founded on the methods the models use to treat inputs: methods in the spatial domain, whose input is transformed into a two-dimensional image, which is ready to work with CNN for sequence feature extraction [39,67]; methods in the temporal domain, for which the input is treated as a word, and works perfectly with RNN [34]. Then, in December 2021, Xiao et al. proposed an ensemble learning model that uses both the spatial and temporal domains to extract the necessary sequence features, in addition to an “attention mechanism” to give interpretability. The on-target model, which is named **AttCRISPR**, was further enhanced with hand-crafted biological features, outperforming even the DeepHF tool with its training dataset. A Python code is provided for further implementation (https://github.com/South-Walker/AttCRISPR; accessed on 30 April 2023).

Lately, almost all gRNA design tools have turned their vision to implement only DNN or hybrid models. These models are increasingly perfecting the predictive activity, becoming more and more computationally flawless. In 2022, Zhang et al. [25] launched the off-target **CRISPR-IP** tool, which includes four layers, each of which performs distinct procedures focused on characterizing novel sequence features; between these layers are CNN, Bi-directional Long-Short Term Memory (BiLSTM, an RNN derivative), an attention layer, and finally a dense layer. The model uses as a training dataset experimental information based on sequencing (SITE-seq and CIRCLE-seq). Finally, epigenetic information and bulge treatment were adapted to the model, resulting in high predicting performance. Regarding the most recent on-target prediction tool, Li et al. [77] proposed a machine learning and deep learning hybrid model. They drew inspiration from a fully computational approach published by Ren et al. [98] that seeks to provide an accurate and high-performance image classification based on XGBoost (extreme gradient-boosted tree, being the machine learning part) and CNN (the deep learning and feature extraction part). The computational approach thus was fused with the biological vision in the hybrid model named **CNN-XG**, using as input a gRNA sequence, treating it with the CNN layer for feature extraction, and finally sending the latter as an input for the XGBoost classification structure. Neither the CRISPR-IP tool nor the CNN-XG tool provide web-based tools, but Python code is available (https://github.com/BioinfoVirgo/CRISPR-IP, https://github.com/MoonLBH/CNN-XG; accessed on 30 April 2023), in which users can train these models with personalized data.

## 6. Conclusions and Future Directions

In recent years, computational approaches have played a pivotal role in the design of highly accurate gRNAs for CRISPR/Cas9 systems. The increasing utilization of CRISPR/Cas9 for gene editing has sparked the development of novel computational tools to mitigate off-target effects and improve the precision of gene modifications. From the early stages of non-learning algorithms to the more recent advancements in complex multi-layer or hybrid machine learning and deep learning architectures, a diverse range of computational and biological factors have been explored to enhance the efficiency and effectiveness of gRNA design.

DNNs have demonstrated remarkable performance in gRNA prediction; however, their limited interpretability challenges the understanding of the underlying biological mechanisms. As a result, researchers have turned their attention back to machine learning models, seeking a balance between performance and interpretability. Notably, in 2020, the introduction of CRISPRpred(SEQ) showcased the resurgence of machine learning models in gRNA design, offering users a powerful and intuitive tool for gRNA prediction [75]. This renewed focus on machine learning models has prompted the integration of machine learning structures into the architecture of DNNs, resulting in hybrid frameworks that harness the strengths of both approaches. CNN-XG, a notable example of such a hybrid model, has demonstrated promising results, surpassing previous architectures while retaining the desirable features of learning-based algorithms [77]. Consequently, the assembly of hybrid models including approaches from DNN and machine learning must be investigated with a significant potential for advancing gRNA design.

The challenges for future research in the design of gRNA are indeed enormous. Looking ahead, future research in gRNA design presents both challenges and opportunities. One important area of focus is the optimization and fine-tuning of hyperparameters in the architecture design to enhance the interpretability of the models. The ability to extract meaningful insights and understand the decision-making process of the models is of paramount importance for researchers and biologists working with CRISPR/Cas9 systems. Furthermore, the integration of biological features observed in laboratory experiments and in silico simulations is essential for bridging the gap between computational predictions and biological reality. Incorporating known biological constraints and epigenetic considerations into the models will enable more accurate and context-specific gRNA design.

To advance gRNA design and improve prediction accuracy, the availability of comprehensive and diverse CRISPR/Cas9 activity datasets is critical. Acquiring data from various human and plant cell lines, as well as unicellular organisms, will provide a broader understanding of the biological factors influencing gRNA activity and efficacy. The accumulation of such data will facilitate the identification of additional biological functional features, enabling the development of more robust and adaptable computational models.

## Figures and Tables

**Figure 1 biomolecules-13-01698-f001:**
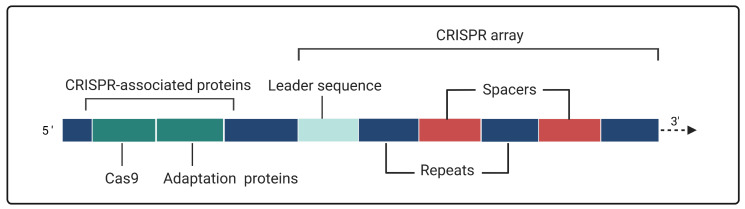
The CRISPR cluster. The CRISPR-associated (Cas) cluster consists of genes coding for the needed proteins for the correct function of the system, such as Cas9, or adaptation-involved proteins (Cas1, Cas2, etc.). The leader sequence controls the expression of the CRISPR array when it becomes necessary; spacers are non-repetitive sequences that separate the repeats; and repeats are strongly conserved palindromic sequences capable of hairpin formation.

**Figure 2 biomolecules-13-01698-f002:**
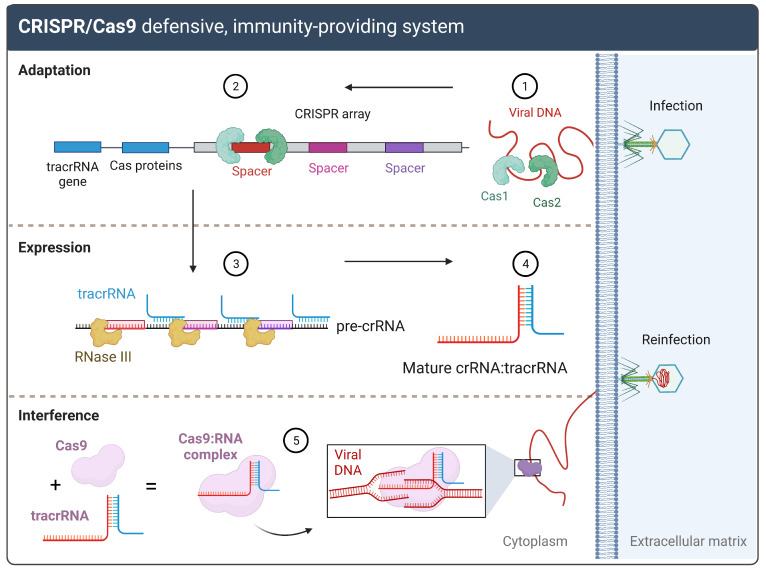
Immunity acquisition during viral infection. (1) The Cas1-Cas2 multimeric complex recognizes and cuts a specific sequence in the viral DNA known as a “protospacer”. (2) Sequentially, the complex integrates the protospacer upstream of the CRISPR array, exactly next to the leader sequence. (3) When the bacteriophage reinfects, the leader sequence initiates the CRISPR array expression, which yields a pre-crRNA. Then, it undergoes selective ribonucleotide sequence degradation by RNase III with parallel binding of the tracrRNA to the desired sequence, (4) generating the mature crRNA:tracrRNA complex. (5) The latter complex binds to the Cas9 protein, which digs into the viral DNA to target the complementary sequence to the crRNA for its cleavage.

**Figure 3 biomolecules-13-01698-f003:**
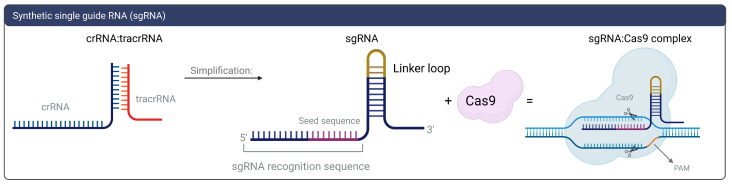
Single guide RNA. Rather than possessing a double RNA (crRNA and tracrRNA), a single RNA is synthesized using a linker loop, yielding the single guide RNA. The protospacer-adjacent motif (PAM) is composed of three to six nucleotides that are recognized by the Cas9 protein.

**Figure 4 biomolecules-13-01698-f004:**
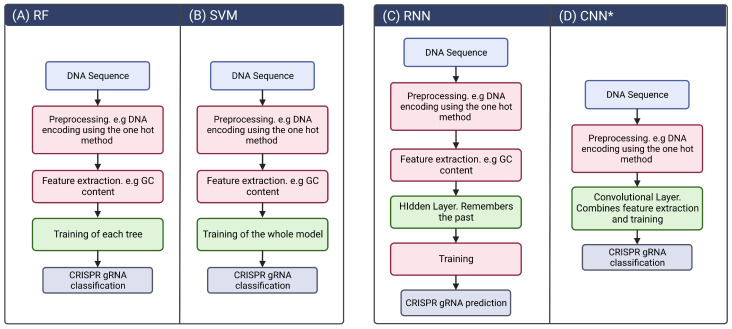
Some of the CRISPR prediction machine learning algorithms. (**A**) Random Forest creates multiple decision trees on different subsets of the dataset and combines their predictions to obtain the final output. (**B**) Support Vector Machine (SVM) separates RNA sequences into two classes that can be non-linear. (**C**) In the RNN architecture, hidden layers are fed back as input to the next layer, allowing the network to capture the temporal dependencies between nucleotides. (**D**) CNN uses a series of convolutional layers to extract features and learn from the DNA sequence. The convolutional layers detect specific patterns. * The CNN architecture is the only architecture capable of dealing with images.

**Figure 5 biomolecules-13-01698-f005:**
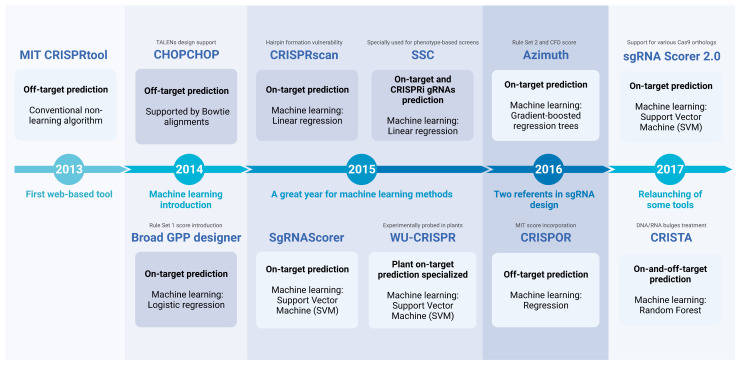
Timeline from 2013 to 2017. This first stage was marked by the implementation of conventional algorithms, and the emergence of machine learning-based algorithms in the tools. Many of these relied heavily on datasets published by others for the training stage. As the field progressed, these tools began to acknowledge their limitations and released upgraded versions targetting the identified problems, resulting in significant improvements.

**Figure 6 biomolecules-13-01698-f006:**
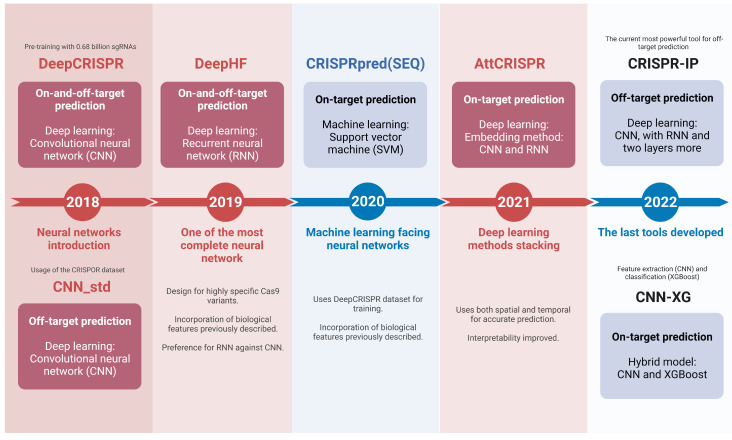
Timeline from 2018 to 2022. With the launch of DeepCRISPR, deep neural networks initiated their treasure, improving each year with the introduction of CNNs RNNs, embedding methods, hybrid models, or the addition of more layers.

**Table 1 biomolecules-13-01698-t001:** Off-target models.

Name	Model	Year	Evaluation Metric	Detail	Reference
CRISPRtool/MIT Design	Conventional non-learning	2013	NA	The first tool for gRNA design.	[22]
CRISPOR	Self-assembled algorithm	2016	Spearman: 0.71–0.77. AUROC: 0.91	Datasets of gRNA sequences and their off-targets from different studies.	[66]
CRISTA	RF	2017	Spearman: 0.81. AUROC: 0.96. AUPRC: 0.96	Assembled dataset for training and validation: GUIDE-Seq, HTGTS, and BLESS.	[48]
Predict CRISPR	SVM	2018	AUROC: 0.99. AUPRC: 0.45	One-hot encoding over the Haeussler dataset.	[55]
Elevation	DT	2018	Spearman: 0.98	One-hot encoding over GUIDE-seq. Boench V2 and Haeussler.	[49]
DeepCRISPR	CNN	2018	Spearman: 0.246. AUROC: 0.804, AUPRC: 0.303	One-hot encoding over GUIDE-seq data.	[67]
CNN_std	CNN	2018	AUROC: 0.972	One-hot encoding over CRISPOR dataset and GUIDE-seq dataset.	[39]
SynergizingCRISPR	FNN, RF, SVM, DT	2019	Spearman: 0.938. AUPRC: 0.299	GUIDE-Seq and Haeussler dataset.	[61]
CHANGE-seq	DT	2020	AUROC: 0.995. AUPRC: 0.881	One-hot encoding.	[50]
CRISPcut	LG, RF, DT.	2020	Accuracy: 0.9149. AUROC: 0.97	One-hot encoding over CIRCLE-seq and CRISPcup.	[68]
CRISPR-Net	RNN-CNN	2020	AUROC: 0.969. AUPRC: 0.477	One-hot encoding over CIRCLE-Seq and GUIDE-seq datasets.	[69]
R-CRISPER	RNN	2021	AUROC: 0.991. AUPRC: 0.319	One-hot encoding over CIRCLE-Seq, SITE and GUIDE datasets.	[26]
piCRISPR	RNN-CNN	2021	AUROC: 0.995. AUPRC: 0.725	One-hot encoding over crisprSQL dataset.	[65]
CROTON	CNN	2021	AUROC: 0.94	One-hot encoding over FORECasT and SPROUT datasets.	[70]
AttCRISPR	Embedding method	2021	Spearman: 0.872	One-hot encoding over DeepHF dataset.	[62]
CRISPR-IP	CNN	2022	Accuracy: 0.990. AUROC: 0.982. AUPRC: 0.751	CIRCLE-Seq dataset and SITE-Seq dataset.	[25]

**Table 2 biomolecules-13-01698-t002:** On-target models.

Name	Model	Year	Evaluation Metric	Detail	Reference
Broad GPP	LG	2014	Spearman: 0.87	One-hot encoding over 1841 gRNAs.	[29]
WU-CRISPR	SVM	2015	AUROC 0.91. Spearman 0.70	Doench and Chari datasets.	[51]
SSC	LG	2015	AUROC : 0.729	Wang, Koike-Yusa, Shalcm, Zhou, Gilbert and Konermann datasets.	[46]
CRISPRScan	LR	2015	Spearman: 0.68	1280 gRNAs in the zebrafish genome.	[44]
SgRNAScorer	SVM	2015	Accuracy: 0.737. AUPRC: 0.758	SpCas9 and St1Cas9 datasets.	[52]
Azimuth	SVM, LG	2016	AUROC: 0.80	One-hot encoding over Avana and GeCKO libraries.	[53]
ge-CRISPR	SVM	2016	AUROC: 0.54–0.93	sgRNAdesigner, CRISPRScan and sgRNAscorer datasets.	[54]
CRISPRater	LR	2017	Spearman 0.67	Wang, Koike-Yusa and Xu datasets.	[45]
SgRNAScorer 2.0	SVM	2017	Accuracy: 0.737. AUPRC: 0.758	SpCas9 and St1Cas libraries.	[71]
CRISPRpred	SVM	2017	AUROC: 0.85. AUPRC: 0.56	K-mer encoding over Broad GPP.	[72]
DeepCRISPR	CNN	2018	AUROC: 0.981. AUPR: 0.497. Spearman: 0.406	One-hot encoding over 15,000 gRNAs from four different cell lines.	[67]
DeepCpf1	CNN	2018	Spearman: 0.87. AUROC: 0.89	One-hot encoding over different datasets oncluding Kleinstiver, Chari and Kim.	[73]
TUSCAN	RF	2018	Spearman: 0.8. AUC of 0.63	Chari, Doench, Horlbeck and Moreno-Mateos databases.	[74]
DeepHF	RNN	2019	Spearman: 0.867	One-hot encoding over ten public datasets.	[34]
CRISPRpred(SEQ)	SVM	2020	Spearman: 0.829. AUROC: 0.893	Haeussler and DeepHF datasets.	[75]
GNL-Scorer	FNN	2020	Spearman: 0.502	One-hot encoding over ten public datasets.	[76]
C-RNN CRISPR	CNN-RNN	2020	Spearman: 0.877. AUROC: 0.976	One-hot encoding over Chuai dataset.	[58]
On-target CRISPRon	CNN	2021	Spearman 0.91	One-hot encoding over 12 K dataset.	[63]
CNN-XG	CNN-Tree	2022	Spearman 0.7352 AUROC: 0.992	Ten public datasets.	[77]

## Data Availability

No new data were created.

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
