# Peer review of "gRNA Design: How Its Evolution Impacted on CRISPR/Cas9 Systems Refinement"

_biomolecules, 2023, doi:10.3390/biom13121698_

Round 1
Reviewer 1 Report
This manuscript reviewed the progress in CRISPR/Cas9 guide RNA (gRNA) design, with a focus on the machine learning (including deep learning) based computational approaches. The authors also introduced of the mechanism of CRISPR/Cas9 systems, and summarized the design principles that are used or discovered in these computational approaches. The manuscript is clear organized and well written. As a review paper, it covers a variety of computational models, and discussed the advantages and disadvantages. As being said, some issues need to be fixed:
1. Figure 5: firstly, the captions and the subfigures are not matched; the left subfigure is CNN, but it is explained as RNN is the caption. Second, the visualization of RNN is wrong. What is shown as RNN in the manuscript is actually fully-connected network.
2. Many abbreviations are not spelled out at the first time they appear. E.g., what is DCDNN and GBRT in Tab. 1, and what is BWA in line 320?
3. Tabs. 1 & 2:
a. the name of the column “Parameter” is wrong. It should be accuracy.
b. these accuracies are based on different test sets, which are not comparable and thus misleading.
c. In addition, it is helpful to include the training dataset for each model.
d. The first one in Tab. 1 has wrong name (“Experiment”) and Model (“gRNA Optimization”)
4. Line 311: it is helpful to have discussion or explanation of “Rule Set 2, and the CFD score”.
5. Line 215: typo “a RNN are…”
Reviewer 2 Report
First, I have to notice that the authors of this manuscript do not seem to be experts in the CRISPR/Cas9 field. At least, I was not able to find a single paper published by the authors and dedicated to the subject. This is not an issue by itself, but it is a warning sign.
Second, I am a "wet" biologist, who does routinely use the CRISPR/Cas9 approach for different purposes. Therefore, I judge the manuscript primarily from this point of view.
Below I summarize only my MAJOR comments.
Overall, the manuscript looks like just as a collection of some facts, most of which are just the names of different approaches. Almost nothing is explained to readers. Instead, many things are just conceptually mentioned. For example, Figures 4 and 5 do not provide any useful details to people like me, who are not familiar with the discussed methods. The accompanying text also does not help. Another example is the organization of the Tables. With rare exceptions, the abbreviations and parameters mentioned in them are not explained at all. What are they? Why are they important?
Many fragments of the text almost completely lack references, for example in the section “3. Neural Networks in gRNA Design” there is only one reference, although the subject is very complicated. This is not acceptable; each statement should be supported by the appropriate reference(s).
In addition, the authors make too much emphasis on the negative sides of the approaches mentioned. It is obvious that nothing is ideal and the algorithms will be further optimized for a long time. Therefore, it would be nicer to emphasize on the advances first. Moreover, since there are many different reviews on the similar subject, the authors could spend MUCH MORE efforts on describing the things, which are critical for understanding the approaches they discuss in detail. How exactly the approaches work and are used in practice? What kind of data and how exactly should be prepared to run each pipeline? Just common words mentioning one or another algorithm is not enough. At least, this is certainly does not seem to be useful for the “wet” community.
Authors completely ignore approaches, which are indeed popular among wet biologist, e.g. CHOPCHOP (https://chopchop.cbu.uib.no/).
In addition to Cas9-based systems, it would be beneficial to also describe nuances of designing gRNAs for other Cas proteins, e.g. Cas12 and Cas13.
Finally, the Introduction of the manuscript also has to be improved substantially. Currently, it suffers from the presence of some nonessential or not so much important facts (e.g., TALENs and ZFNs, the entire Figure 3, the lower part of Figure 1), low quality of data presentation (top of Figure 1 and entire Figure 2), and absence of strong logic through the text.
Spelling mistakes, the usage of rare/fancy words.
Round 2
Reviewer 2 Report
I think that the current version of manuscript can be accepted for publication after the following minor changes are introduced in the text.
Line 28: The meaning of the word “domains” is not clear/obvious. Rephrasing might help here.
Line 49: It is not clear what is meant under “CRISPR locus” here. Changing to “CRISPR array” would make the reading more fluent.
Line 50: Consider changing “degraded” to “processed”, as this is rather a controlled process.
Lines 53-54: It is worthwhile schematically showing the location of the PAM in Figure 2 or/and Figure 3.
Figure 2 legend: A dot is missing after the title of the Figure (“Immunity acquisition during viral infection”).
Lines 77-80: I still find such kind of description as uninformative. Much more details and explanations about different classifications of off-target gRNA binding should be provided. Particularly, what are three types proposed by Manghwar, Zhang, and Niu? In addition, what are two more general and simplified types proposed by Borrelli et al.?
Lines 93-95: The sentence is very difficult for understanding. Rephrasing/simplification might help here. Particularly, what are “computational areas”?
Line 106: First, what is “the sequence of the CRISPR locus”? It seems that the bacterial CRISPR locus is meant here, although I doubt it. Second, is genetic context of THE TARGET LOCUS is meant under “genetic context”?
Line 162: The extra dot immediately before “[36]” should be deleted.
Line 165: A dot is missing at the end of the sentence.
Table 1 and Table 2: It is not clear where descriptions provided in the column “Detail” end and start for each Model.
Figure 2 legend, 2nd line from the bottom: The extra space after “patterns” should be deleted.
Line 361: A reference is required at the end of the sentence ending with “whereas the lowest one positively did”.
Line 368: Should “apport” be changed to “support”?
Line 364: The gene name (“CD45”) should be written in italic, according to gene nomenclature (see details, for example, here: https://en.wikipedia.org/wiki/Gene_nomenclature).
Line 368: A dot is missing at the end of the sentence.
Line 373: Should “gradient-bossed” be changed to “gradient-boosted”?
Line 381: A web-link is required after “only Python code is available”.
Line 464: A space should be added after “[79]”.
Lines 472-473: “Chari et al.” does not seem to be connected with the rest of the sentence.
Line 517: A dot is missing at the end of the sentence.
Line 553: A dot is missing after “[92]”.
Line 557: A dot is missing after “[94]”.
Minor editing of English language is required.
Author Response
Please see the attachmen
